# Transmission, Pathological and Clinical Manifestations of Highly Pathogenic Avian Influenza A Virus in Mammals with Emphasis on H5N1 Clade 2.3.4.4b

**DOI:** 10.3390/v17121548

**Published:** 2025-11-26

**Authors:** Sandra Vibeke Larsen, Rebekka Israelson, Charlotte Torp, Lars Erik Larsen, Henrik Elvang Jensen, Charlotte Kristensen

**Affiliations:** Department of Veterinary and Animal Sciences, University of Copenhagen, 1870 Frederiksberg C, Denmark

**Keywords:** influenza A (H5N1), highly pathogenic avian influenza virus, clade 2.3.4.4b, pathogenesis, avian influenza, mammals, zoonoses, review

## Abstract

Highly pathogenic avian influenza A virus (HPAIV) H5N1, clade 2.3.4.4b, has emerged as a significant zoonotic threat. H5N1 is widely circulating in wild birds, and an increasing number of spillover events have been observed in a wide range of mammalian species. These cases are primarily reported in countries on the European and American continents. This review describes the likely transmission routes, lesions, and clinical manifestations of HPAIV H5N1 clade 2.3.4.4b in naturally infected mammals, with a focus on the involvement of the central nervous system (CNS). In the analysis, pathological findings were categorized by organ system and host species, which were further divided into terrestrial mammals, marine mammals, and dairy cattle. The most frequently reported clinical manifestations were neurological and respiratory signs in marine mammals and neurological signs and lethargy in terrestrial mammals. Macroscopic and histological lesions were commonly found in the CNS and lungs of terrestrial and marine mammals, while dairy cattle showed mainly gastrointestinal and mammary gland involvement. Immunohistochemistry and reverse transcriptase real-time PCR analyses confirmed high viral loads in brain tissues, indicating a neurological tropism of H5N1 clade 2.3.4.4b. Routes of CNS invasion remain uncertain, though both hematogenous and olfactory nerve pathways are discussed. Recent evidence suggests mammal-to-mammal and vertical transmission, raising concerns for the zoonotic and pandemic potential of this virus. In conclusion, the findings emphasize an urgent need for enhanced surveillance to effectively disclose changes in viral pathogenicity and transmissibility among mammalian hosts.

## 1. Introduction

Avian influenza A virus (AIV) belongs to the enveloped virus family Orthomyxoviridae which is a group of linear negative-sense, single-stranded RNA viruses. The virus family contains nine different genera, including *Alphainfluenzavirus,* with the species *alphainfluenzavirus influenzae* (formerly known as influenza A virus (IAV)) as the most relevant member within a one-health perspective because of the widespread circulation in humans, several different mammals, and birds [1,2].

The genome of IAV consists of eight different gene segments that encode at least 12 viral proteins [3]. As for all RNA viruses, the mutation rate is high for IAV due to the lack of proofreading and repair activity of the RNA-dependent RNA polymerase (RdRp) [4]. The genes coding for the surface proteins, hemagglutinin (HA) and neuraminidase (NA), are subject to a high mutation rate, as these proteins are major targets for the host immune system [5]. These mutations give rise to several subtypes, which are classified according to the genomic sequence of the HA and NA genes (e.g., H5N1). The HA protein is responsible for binding to the host receptors, sialic acid (SA) on the host cell surface and mediates the fusion of IAV and host endosomal membrane, allowing entry of the virus into the host cell. The NA protein removes decoy receptors present in mucus and enables the release of new virions from the host cell by cleaving the SA [1,3]. While AIVs preferentially bind to α2,3-linked SA receptors on the host cells, mammalian IAVs prefer an α2,6 linkage [3,6].

AIV is further classified according to its pathogenicity in poultry. Low pathogenic strains (LPAIV) cause local and often subclinical infection, while highly pathogenic strains (HPAIV) may cause systemic infection characterized by high morbidity and mortality in affected herds [7]. In waterfowl, HPAIV typically causes subclinical disease [1], but the recent H5N1 clade 2.3.4.4b has caused significant morbidity and mortality in a wide range of wild birds, including ducks [8]. The major difference between LPAIV and HPAIV is mediated by two classes of proteases, expressed by different body tissues. The trypsin-like proteases that cleave LPAIV HA glycoproteins are restricted to epithelial cells. Therefore, LPAIV is restricted to infecting epithelial-containing tissues found primarily in the respiratory and intestinal tract. In contrast, HPAIV HA glycoproteins can also utilize furin and subtilisin-like proteases for cleavage. These additional proteases are widely distributed, facilitating infection and lesions in several visceral organs, such as the cardiovascular system, but also the central nervous system (CNS) [9]. Conversion of LPAIV to HPAIV can occur by substitution or insertion of basic amino acids in the proteolytic cleavage site (PCS) of the HA glycoprotein, leading to enhanced cleavability [10].

The HPAIV H5N1 subtype circulating today can be traced back to the A/goose/Guangdong/1/96 strain identified among geese in southeast China in 1996 [11]. Since 2003, different lineages of this subtype have spread across continents in spatiotemporal waves, with the most recent being driven by clade 2.3.4.4b [11]. HPAIV H5N1 poses an increasing threat to human and animal health. According to EFSA (European Food Safety Authority), the spread of H5N1 clade 2.3.4.4b from Europe to the American continents during the last two years has led to an intensified HPAIV situation. Recently, clade 2.3.4.4b was detected in infections of mammals, with clinical signs varying from nonspecific illness in domesticated dairy cattle in the USA [12] to mass death among pinnipeds along the South American coastline [13]. The influenza risk assessment tool (IRAT) scores the virus in the “moderate risk” category in terms of potential future emergence and public health impact [14].

In April 2023, a human case of H5N1 was reported in a 53-year-old man in Chile—the genomic sequencing of the virus showed >99.9% identity with H5N1 sequences from Chilean birds, indicating an avian spillover event [15]. From January 2022 to June 2024, 29 isolated human cases have been documented from nine countries, with no signs of transmission between humans [16]. In 2025, the spillover frequency increased, as 26 cases were documented between January and August from 8 different countries [17]. Multiple infections resulted in death, and all cases were confirmed or suspected to have had direct contact with poultry and/or wild birds [11].

The pathogenesis of H5N1 clade 2.3.4.4b remains to be completely understood. Although experimental studies have shed light on how the virus may behave in animal models, these findings may not accurately reflect the lesions observed in naturally infected mammals. Of particular concern, frequent reports of neurological signs and neurological lesions in infected mammals are often followed by fatal outcomes. The mechanism by which H5N1 clade 2.3.4.4b infects mammals and disseminates to the CNS remains unclear. The aim of this review is to compile existing literature on H5N1 clade 2.3.4.4b infections in mammals, with a focus on transmission and clinical and pathological manifestations.

## 2. Materials and Methods

This review is based on two literature searches in the database PubMed and Web of Science, using the search words: (((mammal*) AND (((Patho*)) OR (clinical))) AND ((((HPAIV) OR (high* pathogenic avian influenza virus)) AND (H5N1)) AND (2.3.4.4b))) NOT (((Experimental) OR (diagnostic)) OR (review)). The searches were completed without a restriction on the publication date, due to the limited literature and current development of the topic. This study focuses on the pathogenesis of HPAIV, subtype H5N1, clade 2.3.4.4b, in mammals, and our search was limited to this specific clade by these keywords. By excluding the words “experimental” and “diagnostic” from the main search, the results were limited to naturally infected cases. Lastly, only primary articles were included and reviews were excluded. Searches were completed on 29 April 2025, resulting in 102 articles from PubMed, and on 15 November 2025, resulting in 583 articles from Web of Science. The PRISMA flow diagram (Figure 1) illustrates the sorting process of the articles.

The 27 included articles were grouped into three categories based on the species reported:-Terrestrial mammals (including domesticated cats)-Marine mammals-Dairy cattle

Macroscopic lesions were divided into different groups according to the affected organ systems for both terrestrial and marine mammals. The pathological findings were first categorized by their location in the host rather than the type of lesion. For example, hemorrhage in the cerebrum would be categorized as a “CNS lesion”, even though it is a cardiovascular condition. The “cardiovascular lesions” include all lesions found in relation to the heart or diffuse vascular lesions (e.g., subcutaneous hemorrhage). When counting lesions, several lesions in the same organ system were only counted once. Therefore, an animal that presents both congestion and edema in the lungs was only counted as one finding in the group of “respiratory lesions”.

Articles included were required to report PCR-positive cases of infection with HPAIV H5N1 clade 2.3.4.4b (from now on referred to as H5N1), thereby excluding pathological findings in mammals that were unrelated to this specific clade. Furthermore, articles only describing phylogenetic relationships between subtypes were excluded. Several of the species groups include animals found in many different countries. Only animals with necropsy data were included in the analysis. Thus, the number of cases included does not represent the actual number of reported cases of H5N1 infections in mammals. Lastly, the articles describing respiratory lesions from drowning were excluded.

## 3. Transmission, Clinical and Pathological Manifestations Among Mammals

### 3.1. Introduction to Papers

H5N1 has been documented in many different mammalian species, and this analysis categorizes the species into two main groups defined as “terrestrial mammals” and “marine mammals”. H5N1 has also been detected in dairy cattle, discussed separately as the lesions found in these animals differ from other mammals. Figure 2 shows an overview of the number of terrestrial mammals (*N* = 130) (Appendix A) and marine mammals (*N* = 58) included in this analysis (Appendix A).

The red fox is the most frequent species in the group of terrestrial mammals, followed by coatis and cats (Figure 2A). The sea lion is the most frequent species in the group of marine mammals, followed by the harbor seal and the elephant seal (Figure 2B).

Necropsy data from six infected cows was obtained from Burrough et al. 2024 [12].

Geographical distribution of mammals included in this review (Appendix A). Most mammals included in this review were from America, primarily North America. Many countries and continents have not reported cases of mammals where pathology was included as shown by Figure 3A. The distribution of different species is shown in Figure 3B–D.

### 3.2. Transmission Routes

Three main transmission routes have been described for H5N1 in mammals, which include sporadic spillovers from wild birds, horizontal transmission between mammals, and transplacental transmission. The transmission routes have been defined by epidemiological and phylogenetic analyses based on the whole virus genome sequencing [13]. The transmission routes are summarized in Figure 4.

#### 3.2.1. Spillover from Birds

The ingestion of birds infected with H5N1 was the most frequently observed or hypothesized transmission route for infection in mammals [18,19,20,21,22,23,24,25]. Infection in wild terrestrial predators is often associated with ingestion of waterfowl, which is common for many species such as foxes, bobcats, and coyotes [18,19,25]. For domestic cats, infections were associated with the feeding of raw meat [26]. Different marine animals, such as harbor seals, have been infected, although they do not typically prey on or scavenge birds [27,28,29]. In addition, it has been hypothesized that indirect transmission through environmental exposure, including accidental ingestion of feces or feathers, is a potential transmission route to mammals [28,30].

The exact initial source of infection of American dairy cattle remains unknown, but the virus probably originated from infected wild birds [12].

#### 3.2.2. Mammal-to-Mammal Transmission

Until recently, transmission of H5N1 between mammals was not documented. However, since 2022, more cases of mammal-to-mammal infections have been suspected, in connection with an outbreak where the virus killed thousands of marine mammals along the South American coast [31,32]. Elephant seals were frequently affected, and a mortality rate of >95% of the pups was observed. This mass infection could not be explained by sporadic spillovers from seabirds, as the pups’ interaction with seabirds was limited. The fact that the pups exclusively nurse on milk from their mothers supports the suspicion of vertical transmission [13]. Shortly after, outbreaks were observed in American dairy cattle, with horizontal transmission being the most likely route of infection, since disease spread to dairy farms across many different states after they received cows from infected regions. Cross-species transmission from cows to cats was also suspected, as H5N1-positive cats were found dead in the affected dairy cattle herds. The cats most likely became infected after ingestion of contaminated, unpasteurized milk from the infected cows [12]. Furthermore, a detailed phylogenetic analysis of the H5N1 mass outbreak in South American pinnipeds during 2022–2023 indicated mammal-to-mammal transmission [8].

Lastly, transplacental transmission was suspected as a transmission route during an outbreak in elephant seal pups in Argentina [13]. Furthermore, a recent study detected H5N1 in the placenta and fetus from a pregnant South American sea lion, which supports the hypothesis of vertical transmission [33].

### 3.3. Clinical Signs

The frequencies of clinical signs associated with H5N1 infection were compared among the 99 terrestrial mammals. The most common clinical signs observed were neurological signs, lethargy, and respiratory signs. A wide range of neurological manifestations, such as increased vocalization, paralysis, circling, convulsions, ataxia, hypermetria, depression, and opisthotonus were observed. The only respiratory sign noted was dyspnea. Figure 5A illustrates the frequencies of clinical signs observed among terrestrial mammals (Appendix A). From a blood analysis, one Tibetan black bear from a zoo showed signs of acute renal failure [24].

When comparing the frequency of clinical signs among 26 marine mammals infected with H5N1, the most common signs were neurological signs, respiratory signs, and lethargy (Figure 5B) (Appendix A).

Clinical signs among dairy cattle were limited to non-specific clinical signs of illness including reduced feed intake, drop in milk production, and dehydration. There were no indications of abnormal behavior or other signs of neurological involvement [12].

### 3.4. Macroscopic Lesions

The frequencies of macroscopic lesions in affected organ systems were compared among 86 terrestrial and 28 marine mammals (Figure 6).

The most common macroscopic lesions found in terrestrial mammals were in the respiratory tract followed by the liver and the CNS (Figure 6A). The lung lesions in terrestrial mammals consisted mainly of pulmonary edema, hemorrhages, congestion, failure to collapse, emphysema, and pleural effusion. The liver lesions included hemorrhages, congestion, enlargement, discoloration and the presence of yellow-white foci. The CNS lesions were mainly recorded as brain hemorrhages, congestion, and cerebral swelling (Appendix A).

In marine mammals, respiratory lesions were also the most common finding followed by CNS lesions and lymph adenomegaly (Figure 6B). The respiratory lesions consisted of hyperemic lungs, atelectasis, emphysema, edema, congestion, failure to collapse, dark red discoloration, and diffuse firmness. CNS lesions were dominated by hemorrhages, congestion, and hyperemic vessels (Appendix A). One polar bear showed ulcerative skin lesions around the eye and oral commissure [34].

In contrast, lesions in domesticated dairy cattle differed from those of other mammals, as the GI-tract and mammary glands were the only affected organs. The lesions consisted of ulcerations and/or erosions in the intestines and firm mammary glands, but some cows showed no lesions (Appendix A).

### 3.5. Histopathological Changes

In the following two sections, tissue localization (including viral quantification) and histopathological changes in terrestrial and marine mammals are treated as one group, as these findings were more comparable than the macroscopic lesions.

The frequencies of histopathological changes in affected systems were compared among 84 terrestrial and 24 marine mammals (Figure 7A,B). Histopathological changes were mainly observed in the respiratory and CNS organ systems in both the terrestrial and marine mammals (Figure 7A,B).

The most frequently observed histopathological changes in the CNS were meningoencephalitis associated with neural necrosis, perivascular cuffing with lymphocytes, gliosis, and vasculitis (Figure 8A). These lesions were detected in the cerebrum, cerebellum, truncus encephali and medulla spinalis. In general, the lesions had a multifocal distribution pattern, and the cell infiltrations primarily consisted of lymphocytes and, to a lesser extent, neutrophils (Appendix A).

Interstitial pneumonia, characterized by thickening of the alveolar septa due to infiltration of inflammatory cells, vasculitis, and varying degrees of alveolar exudation, was a common histopathological finding in the lower respiratory tract. In relation to the vascular system, lymphocytes were observed within the walls of blood vessels in several organs, which was diagnosed as vasculitis. These vascular changes were often associated with additional types of lesions in the affected organs. Varying degrees of myocarditis (including cardiomyocyte necrosis), petechial hemorrhage, and hemopericardium were observed. In addition, sporadic cases of hepatitis, splenitis, glomerulonephritis, and adrenocortical necrosis were observed. Depletion of lymphoid tissue was observed in lymph nodes, spleen, thymus, and Peyer patches of the intestines (Appendix A).

No histopathological changes in the gastrointestinal tract were observed among marine mammals, and lesions were only sparsely detected in terrestrial mammals (Appendix A).

Ten bush dogs from the UK exhibited foci of necrosis and monocytic inflammation within and between the inner circular and outer longitudinal smooth muscle layer, associated with capillaries and in the submucosal and myenteric plexus (Auerbach’s plexus) [21].

Chorionic villus atrophy of the placenta was observed in one pregnant sea lion, but no lesions were found in the fetus [33].

Similarly to the macroscopic lesions, none of the above-mentioned lesions were observed in dairy cattle, as the only histopathological changes observed were suppurative mastitis and hepatitis [12].

### 3.6. Tissue Localization and Viral Quantification

Two different methods have been used to confirm infection with IAV in tissue. IAV antigen was detected in tissue samples using immunohistochemistry (IHC) with monoclonal antibodies targeting the IAV nucleoprotein. For molecular detection and quantification, reverse transcription real-time PCR (RT-qPCR) was performed on RNA extracted from affected tissues [19,29], and one study used in situ hybridization targeting the M gene [24]. The most frequent localization of IAV antigen was observed in the brain within neurons, glial cells, and ependymal cells lining the ventricles (Figure 8B, Appendix A). Brain tissue samples had the lowest Ct-values (i.e., high amount of IAV RNA). In one of the three red foxes presenting neurological signs in the Netherlands, IAV antigen was present in both the brain and the olfactory epithelium. No IAV antigen or histopathology was detected in the olfactory bulb of this fox [22]. Positive IAV cells were detected in the myenteric plexus of a Tibetan black bear [24].

Another frequent site of IAV-positive IHC staining was the lung tissue, where antigen was detected in epithelial cells lining the bronchi and bronchioles, as well as pulmonary macrophages. Other sites containing IAV-positive cells included the liver, heart, intestinal tract, pancreas, spleen, kidney, adrenal gland, and retina (Appendix A).

In general, and across mammalian species, lower levels of IAV RNA were detected in tracheo-rectal swabs analyzed by RT-qPCR (higher Ct-values) compared to brain tissues (Appendix A).

One study [33] reported a case of H5N1 in a pregnant sea lion in South America, where IAV antigen was detected in trophoblasts and maternal vessels of the placenta. In the fetus, IAV-positive cells were detected in the lungs, thymus, kidney, and cardiomyocytes.

In domesticated dairy cattle, IAV antigens were only detected in the mammary gland and germinal centers of lymphoid tissue [12]. Thus, no virus was detected in the CNS of cattle.

## 4. Discussion

The detection of IAV antigen in several anatomic locations within the CNS, combined with the pronounced macroscopic and histopathological changes in these organs and the high frequency of neurological signs observed among mammals infected with H5N1, indicates the CNS as a major target for H5N1 in mammals. How the virus gains access to the CNS remains, however, unclear.

The detection of lesions in multiple organ systems, including vasculitis, with concurrent IAV antigen presence, and the observation of lymphoid tissue depletion and inflammation in both secondary and primary lymphoid organs (Appendix A), support a hematogenic dissemination into the CNS. An alternative hypothesis proposes that H5N1 may enter the CNS via the cranial nerves, particularly the olfactory nerve or through the myenteric plexus in the gut. This hypothesis is supported by experimental studies in ferrets [35] and the ability of some H5N1 strains to utilize transaxonal transport [36].

A non-systemic route of transmission could explain the predominance of lesions within the CNS. Initial replication in olfactory neurons might induce localized damage in the olfactory bulb, enabling the virus to reach other CNS regions through cerebrospinal fluid distribution. This theory is supported by the detection of viral antigen in the CNS and olfactory epithelium of one red fox [22].

Even though transmission via the olfactory nerve has been proven in experimental studies, this observation does not necessarily prove that such transmission is possible under natural circumstances. In experimental studies, viruses are inoculated into the nasal cavity at a high concentration, and the animals are kept in a highly controlled environment [35,36]. In contrast, many of the mammals included in this analysis were presumed to have eaten dead birds infected with H5N1 and therefore were orally infected.

To definitively determine if H5N1 invades the CNS via the olfactory nerve in naturally infected mammals, the methods used for sampling and processing CNS tissue require standardization. Many of the articles included do not describe the necropsy process in detail and only present substantial findings. Samples should consistently include a well-defined set of CNS regions, including the olfactory bulb and other anatomical locations along the olfactory nerve pathway. Brain tissue is highly susceptible to autolysis after death, which may affect the changes [37], and there is a risk of contamination from the surroundings when removing the brain from the skull, which makes the sampling more challenging. Additionally, sampling of the superior turbinates—where the olfactory epithelium is located—is difficult due to their anatomical location [37].

Recently, an experimental study was conducted in dairy cattle in which lactating cows were inoculated with H5N1 via the intramammary route, whereas yearling calves were exposed through aerosol inoculation [38]. The pathological changes observed in the mammary glands closely resembled the lesions described under natural conditions [12]. However, minimal multifocal obstructive atelectasis consistent with H5N1 RT-qPCR detection was observed in one out of five calves. These findings suggest that H5N1 can be transmitted between calves via aerosols under experimental conditions. Nevertheless, the limited reports of respiratory involvement in naturally infected dairy cattle question the relevance of aerosol transmission under field conditions. Indeed, a potentially unique virus–host interaction could explain why lesions in infected cattle appear different compared to other mammalian species; however, further studies are needed to elucidate this hypothesis.

Different challenges were met during the processing of data in the analysis of lesions. Firstly, the level of detail in the descriptions of the lesions varies significantly across the different studies, complicating the comparison. Moreover, verification that all the necropsy findings were due to infection with H5N1 was impossible, as the animals were only tested for a few differential etiologies, if any. The respiratory lesions might be underrepresented for the marine mammals, as many were found dead due to drowning (Appendix A). There is a selection bias in the animals represented due to the inclusion criteria and some animal species are more represented than others. Nevertheless, respiratory and CNS involvement was predominant in all species except cattle. Lastly, due to practical constraints, histopathological examination of pinnipeds was not possible excluding a significant group of affected animals from the analysis [13]. Another limitation of the review was the geographic distribution of the available data. Most mammals originated from North or South America with a few cases from Europe and Asia. Especially the lack of data from Asia is concerning, since a large proportion of the human fatalities have been recorded in Asia [39].

The lesions observed among infected mammals may reflect lesions that could occur in humans if the virus acquires the ability to effectively infect and transmit between humans. Reports of human H5N1 infections, with disease severity ranging from conjunctivitis as the only symptom to severe pneumonia and death [31,39], combined with the virus’s classification as a ‘moderate risk’ according to the IRAT analysis, emphasize the urgent need to emphasize its pandemic potential [14]. Additionally, the mammal-to-mammal transmission reported among pinnipeds along the South American coastline increases this pandemic concern. IAV infections typically cause respiratory disease in humans, whereas neurological symptoms are uncommon [40]. Nevertheless, cases of CNS involvement have been reported during influenza infections in humans during the last century, and therefore this feature is not limited to infection with H5N1 [41,42]. The Spanish Flu in 1918, a pandemic caused by the H1N1 subtype, has been linked to neurological diseases such as Guillain-Barré syndrome, Kleine-Levin syndrome, Parkinson’s disease, and neurodegenerative diseases including postencephalitic Parkinson’s disease [41]. Additionally, the olfactory route for infection of the CNS has been confirmed in an immunosuppressed infant with seasonal H3N2 influenza infection [42], but the route to the CNS, in general, remains to be fully understood.

The findings of high levels of virus in various organs of infected mammals underline the importance of the CDC recommendations for using personal protective equipment (PPE) when in direct or close contact with sick or dead animals. Several of the cases in mammals suggest an oral exposure, which further emphasizes that proper cooking of poultry, beef, and eggs is important [43]. The capability of H5N1 to infect such a broad spectrum of mammals supports the updated guidelines for a global one-health surveillance approach, facilitating collaboration between the public health, wildlife, and environmental sectors, provided by the Food and Agriculture Organization of the United Nations (FAO) and the World Organization for Animal Health (WOAH).

## 5. Conclusions

Based on the findings in this review, it is not possible to distinguish conclusively between hematogenous and neural routes of CNS invasion; however, the evidence suggests that both pathways may be important.

CNS and respiratory lesions were both reported with a relatively high frequency among both terrestrial and marine mammals, dominated by interstitial pneumonia and multifocal meningoencephalitis, respectively. This correlated well with the viral loads, with the primary IAV-positive cells being epithelial cells lining the respiratory tract, pulmonary macrophages, and neurons. Collectively, these findings suggest that the pathogenesis of H5N1 in naturally infected mammalian species is comparable and that variation in the route of infection may potentially be responsible for the differences observed in the distribution of lesions.

This study underlines the importance of understanding the pathogenesis of H5N1 in mammals, particularly given the documented cases of mammal-to-mammal transmission. A detailed understanding of how H5N1 clade 2.3.4.4b gains access to the CNS is crucial for risk assessment regarding public health and for guiding authorities in the event of a potential human pandemic.

## Figures and Tables

**Figure 1 viruses-17-01548-f001:**
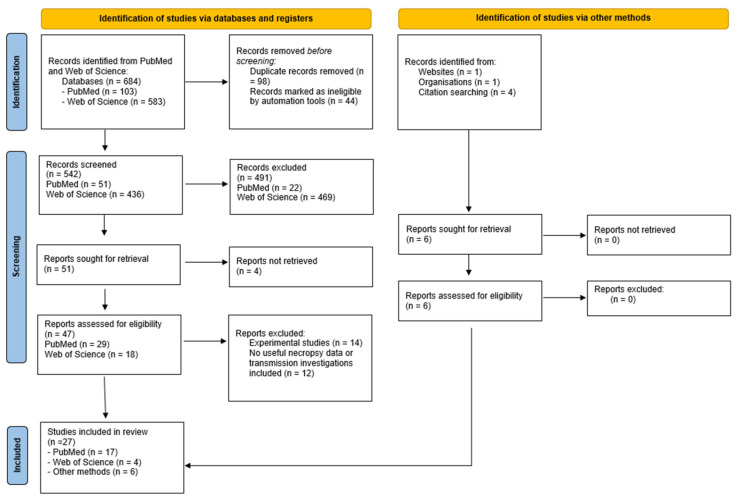
PRISMA flow diagram.

**Figure 2 viruses-17-01548-f002:**
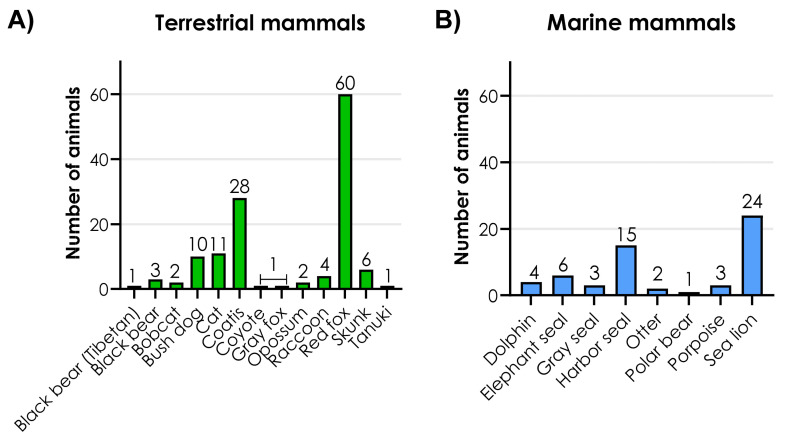
Species frequency of (**A**) terrestrial mammals and (**B**) marine mammals. (**A**) The terrestrial species included: Black bear Tibetan (*Ursus thibetanus*), black bear (*Ursus americanus)*, bobcat (*Lynx rufus*), bush dog (*Speothos venaticus*), cat *(Felis catus*), coatis (*Nasua nasua*), coyote (*Canis latrans*), gray fox (*Urocyon cinereoargenteus*), opossum (*Didelphis virginiana*), raccoon *(Procyon lotor*), red fox (*Vulpes vulpes*), skunk (*Mephitis mephitis*), and tanuki (*Nyctereutes procyonoides albus*). (Appendix A) (**B**) The mammalian species included: Dolphin (*Delphinus delphis*), elephant seal (*Mimrounga leonina*), gray seal (*Halichoerus grypus*), harbor seal (*Phoca vitulina*), otter *(Lontra felina*), polar bear (*Ursus maritimus*), porpoise (*Phocoena phocoena*), and sea lion (*Otaria flavescens*). Source: Appendix A. Created with GraphPad Prism version 10.2.2 for Windows, GraphPad Software, Boston, MA, USA, www.graphpad.com.

**Figure 3 viruses-17-01548-f003:**
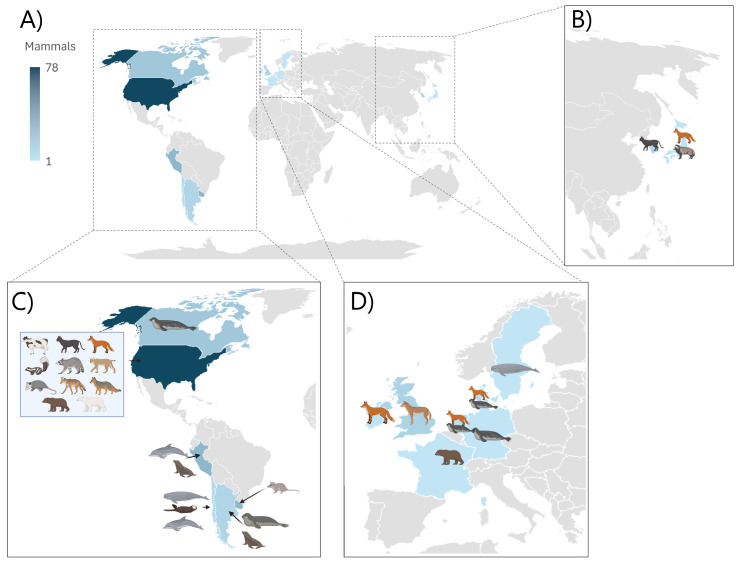
Geographical distribution of mammals. (**A**): Geographical distribution showed by intensity of color from one to seventy-eight animals per country. (**B**): Geographical distribution of species, number of animals is noted in parentheses. South Korea: cat (9), Japan: red fox (1), tanuki (1). (**C**): Geographical distribution of species, number of animals is noted in parentheses. Canada: harbor seal (14), gray seal (1). USA: cow (6), cat (2), red fox (50), striped skunk (6), raccoon (4), bobcat (2), opossum (2), coyote (1), gray fox (1), black bear (3), polar bear (1). Peru: sea lion (21), dolphin (2). Argentina: elephant seal (6), Sea lion (3). Chile: porpoise (2), otter (2), dolphin (2). Uruguay: coatis (23). (**D**): Geographical distribution of species, number of animals is noted in parentheses. United Kingdom: bush dog (10), Ireland: red fox (2), Denmark: red fox (4), harbor seal (1), Sweden: porpoise (1), Netherlands: red fox (3), gray seal (1), France: Tibetan black bear (1), Germany: gray seal (1). Created with Microsoft Excel and in BioRender. Torp, C. (2025) https://BioRender.com/rbek3ml, https://BioRender.com/eh7zvbk. Source: Appendix A.

**Figure 4 viruses-17-01548-f004:**
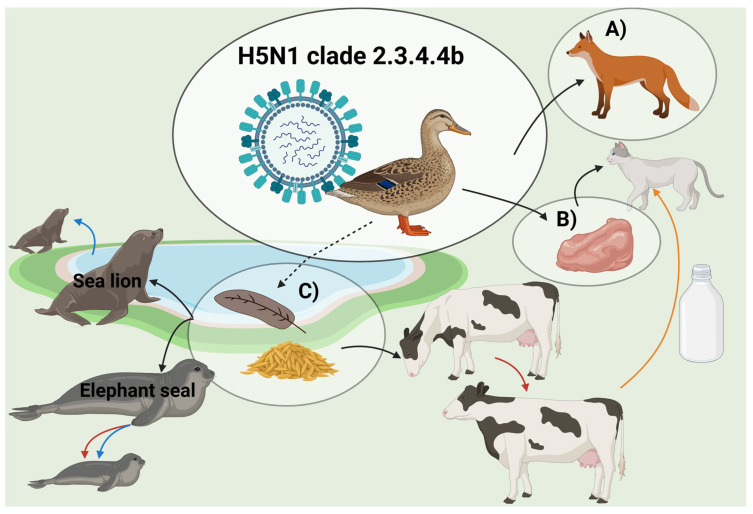
Transmission routes of H5N1 to mammals. Introductions of H5N1 from waterfowl are marked with circles. (**A**) Ingestion of waterfowl by terrestrial predators, (**B**) ingestion of contaminated non-sterilized raw meat by domestic cats, (**C**) indirect contact with contaminated environment of either feces, feathers or water (dash line) to marine mammals and cattle. Red arrows show species with evidenced horizontal transmission and blue arrows show species with hypothesized/evidenced vertical transmission. The orange arrow solid line shows the only mammal cross-species transmission described. Created in BioRender. Kristensen, C. (2025) https://BioRender.com/knqnt4l.

**Figure 5 viruses-17-01548-f005:**
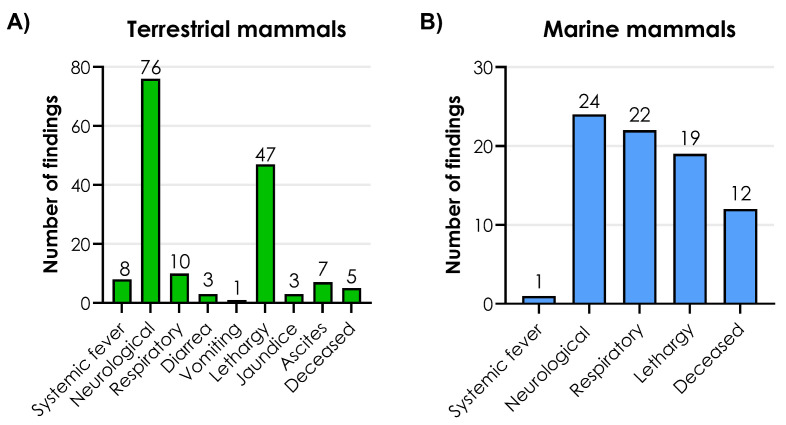
Frequency of clinical signs among (**A**) terrestrial mammals and (**B**) marine mammals. Source: Appendix A. Created with GraphPad Prism version 10.2.2 for Windows, GraphPad Software, Boston, MA, USA, www.graphpad.com.

**Figure 6 viruses-17-01548-f006:**
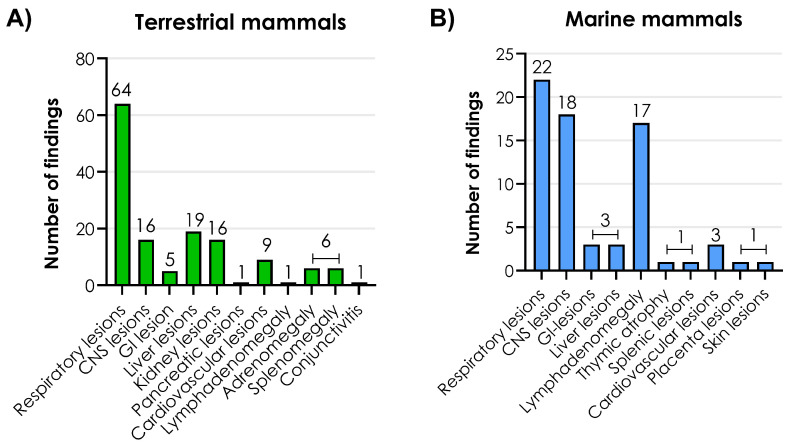
Frequency of macroscopic lesions in (**A**) terrestrial mammals and (**B**) marine mammals. CNS: central nervous system. GI: gastrointestinal. Source: Appendix A—S3. Created with GraphPad Prism version 10.2.2 for Windows, GraphPad Software, Boston, MA, USA, www.graphpad.com.

**Figure 7 viruses-17-01548-f007:**
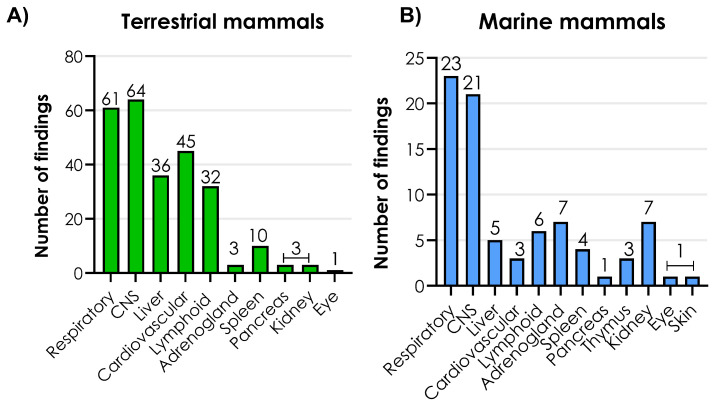
Frequency of histopathological changes in (**A**) terrestrial mammals and (**B**) marine mammals. CNS: central nervous system. Source: Appendix A–S3. Created with GraphPad Prism version 10.2.2 for Windows, GraphPad Software, Boston, MA, USA, www.graphpad.com.

**Figure 8 viruses-17-01548-f008:**
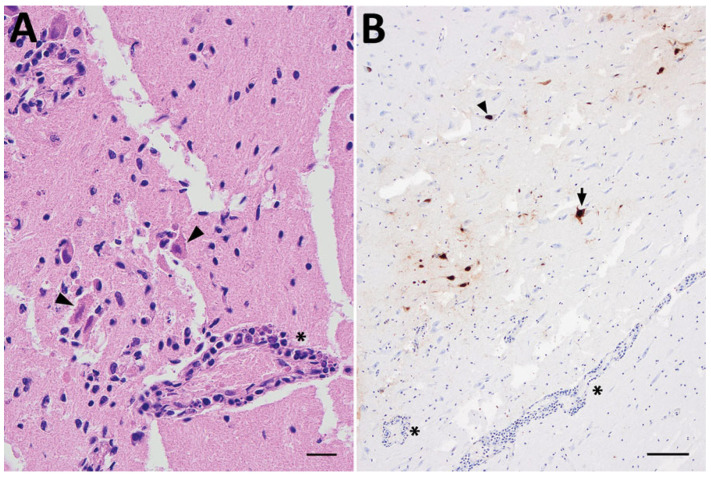
CNS lesions and location of influenza A virus antigen in an infected habour porpoise. (**A**) Histopathological changes in the brain showing neuronal necrosis (arrowheads), perivascular cuffing and vasculitis (asterisk). Scale bar: 20 µm. (**B**) Immunohistochemical staining of influenza A virus nucleoprotein in nuclei of neurons (arrowhead) and cytoplasmic (arrow), perivascular cuffing and vasculitis (asterisk). Scale bar: 20 µm. Figure reprinted from [30].

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
