# Peer review of "Transmission, Pathological and Clinical Manifestations of Highly Pathogenic Avian Influenza A Virus in Mammals with Emphasis on H5N1 Clade 2.3.4.4b"

_viruses, 2025, doi:10.3390/v17121548_

Round 1
Reviewer 1 Report
Comments and Suggestions for Authors
This systematic review comprehensively analyzes H5N1 clade 2.3.4.4b infections in mammals, documenting transmission routes, clinical manifestations, and pathological findings. The study systematically reviewed 22 articles covering 127 terrestrial mammals and 56 marine mammals, plus recent dairy cattle infections. Key findings include neurological tropism with high viral loads in brain tissues across most mammalian species, while dairy cattle show unique gastrointestinal and mammary gland involvement without CNS infection. The authors identify three transmission routes (spillover from birds, mammal-to-mammal, and vertical transmission), analyze CNS invasion mechanisms (hematogenous vs. olfactory routes), and emphasize the public health significance of mammalian adaptation.
- Include the PRISMA flow diagram (Figure 1) that is referenced but missing from the submitted manuscript. Additionally, specify the search fields used in PubMed and consider expanding to additional databases (Web of Science, Scopus) for broader coverage.
- While frequency counts provide descriptive overview, incorporate basic statistical measures (confidence intervals, measures of variability) where feasible, particularly for comparing lesion frequencies between terrestrial and marine mammals.
- Develop more detailed public health recommendations, surveillance strategies, and specific research priorities based on the compiled evidence. Address knowledge gaps more explicitly, particularly regarding the relative importance of different CNS invasion pathways.
- Ensure all referenced figures (Figures 1-7) and supplementary materials (Appendix 1) are properly included in the submission package.
- Provide a more comprehensive discussion of study limitations, including selection bias from excluding animals without necropsy data, reporting heterogeneity across studies, and geographic distribution constraints.
- Expand on the implications of neurological manifestations across species, particularly given the absence of such findings in dairy cattle, which may indicate unique host-virus interactions requiring further investigation.
- Improve transitions between sections and ensure consistent species nomenclature throughout the manuscript.
Author Response
Reviewer 1
This systematic review comprehensively analyzes H5N1 clade 2.3.4.4b infections in mammals, documenting transmission routes, clinical manifestations, and pathological findings. The study systematically reviewed 22 articles covering 127 terrestrial mammals and 56 marine mammals, plus recent dairy cattle infections. Key findings include neurological tropism with high viral loads in brain tissues across most mammalian species, while dairy cattle show unique gastrointestinal and mammary gland involvement without CNS infection. The authors identify three transmission routes (spillover from birds, mammal-to-mammal, and vertical transmission), analyze CNS invasion mechanisms (hematogenous vs. olfactory routes), and emphasize the public health significance of mammalian adaptation.
Thank you for the detailed comments we believe have added significant value to our paper.
- Include the PRISMA flow diagram (Figure 1) that is referenced but missing from the submitted manuscript. Additionally, specify the search fields used in PubMed and consider expanding to additional databases (Web of Science, Scopus) for broader coverage.
The PRISMA flow diagram was included in the submitted manuscript as Figure 1. It has now been updated according to reviewer 2 comments. We have broadened our search to include Web of Science, which resulted in the inclusion of three additional papers.
- While frequency counts provide descriptive overview, incorporate basic statistical measures (confidence intervals, measures of variability) where feasible, particularly for comparing lesion frequencies between terrestrial and marine mammals.
We understand this suggestion, but it will be difficult to apply statistics on this data due to different lesions and no standardized sampling protocols, we do not think this data is appropriate for a statistical analysis.
- Develop more detailed public health recommendations, surveillance strategies, and specific research priorities based on the compiled evidence. Address knowledge gaps more explicitly, particularly regarding the relative importance of different CNS invasion pathways.
Thank you for emphasizing this. We agree that this should be discussed in more detail and have integrated this into the discussion now.
- Ensure all referenced figures (Figures 1-7) and supplementary materials (Appendix 1) are properly included in the submission package.
We have now ensured that all figures are included in our resubmission.
- Provide a more comprehensive discussion of study limitations, including selection bias from excluding animals without necropsy data, reporting heterogeneity across studies, and geographic distribution constraints.
We agree that this is not properly emphasized and have now added these limitations to our discussion and have included a Figure showing geographic distribution.
- Expand on the implications of neurological manifestations across species, particularly given the absence of such findings in dairy cattle, which may indicate unique host-virus interactions requiring further investigation.
We agree that this is important to emphasize and have incorporated it in the discussion.
- Improve transitions between sections and ensure consistent species nomenclature throughout the manuscript.
We have ensured that a consistent species nomenclature is used throughout the manuscript.
Reviewer 2 Report
Comments and Suggestions for Authors
The review article by Larsen and colleagues on transmission, lesions and clinical manifestations found in mammals from searching the literature is both timely and informative. Here they specifically search for articles that describe naturally occurring infections of mammals on which necropsy data are reported. This resulted in 22 articles that were included in the review. Overall, it is well done. I have a few suggestions for improvements to the manuscript.
An updated count of human cases is worth reporting, as is the clinical description of the serious illness and/or death in 2 individuals. The number reported by 2024 is no longer accurate.
Figure 1 final box includes two numbers. One n=17, other n=22. It is not clear what the distinction between them is (and =sign is missing).
In figures 2,4,5 and-6, number is used on the x-axis, however frequency (%) is reported. This is confusing.
Fig. 5A. Y-axis of tissue descriptions are not in English. Also, these figures would be better if tissues were in same order in both A and B.
Minor corrections;
Waterfowl is used for both singular and multiple species.
Line 57 has caused significant
Line 69 among geese
Line 181 Pups exclusively nurse
Line 322 the capacity of transaxonal transport proven among H5N1 (perhaps you mean proven for H5N1 strains of influenza viruses?)
Comments on the Quality of English LanguageSome sentences are cumbersome. While I understand what they mean it does not read easily.
Author Response
Reviewer 2
The review article by Larsen and colleagues on transmission, lesions and clinical manifestations found in mammals from searching the literature is both timely and informative. Here they specifically search for articles that describe naturally occurring infections of mammals on which necropsy data are reported. This resulted in 22 articles that were included in the review. Overall, it is well done. I have a few suggestions for improvements to the manuscript.
Thank you for taking the time to review this paper.
- An updated count of human cases is worth reporting, as is the clinical description of the serious illness and/or death in 2 individuals. The number reported by 2024 is no longer accurate.
We agree and we have now included the number from 2025.
- Figure 1 final box includes two numbers. One n=17, other n=22. It is not clear what the distinction between them is (and =sign is missing).
Thank you for mentioning this – this Figure has now been updated.
- In figures 2,4,5 and-6, number is used on the x-axis, however frequency (%) is reported. This is confusing.
We agree and have changed it to the number of findings instead.
- 5A. Y-axis of tissue descriptions are not in English. Also, these figures would be better if tissues were in same order in both A and B.
Thank you for mentioning this, we have corrected this in the figures.
Minor corrections:
Points 5-9 have all been addressed in the manuscript file now.
- Waterfowl is used for both singular and multiple species.
- Line 57 has caused significant
- Line 69 among geese
- Line 181 Pups exclusively nurse
- Line 322 the capacity of transaxonal transport proven among H5N1 (perhaps you mean proven for H5N1 strains of influenza viruses?)